# Color Visual Illusions: A Statistics-based Computational Model

**Elad Hirsch and Ayellet Tal**
Technion – Israel Institute of Technology
{eladhirsch@campus,ayellet@ee}.technion.ac.il

## Abstract

Visual illusions may be explained by the likelihood of patches in real-world images, as argued by input-driven paradigms in Neuro-Science. However, neither the data nor the tools existed in the past to extensively support these explanations. The era of big data opens a new opportunity to study input-driven approaches. We introduce a tool that computes the likelihood of patches, given a large dataset to learn from. Given this tool, we present a model that supports the approach and explains lightness and color visual illusions in a unified manner. Furthermore, our model generates visual illusions in natural images, by applying the same tool, reversely.

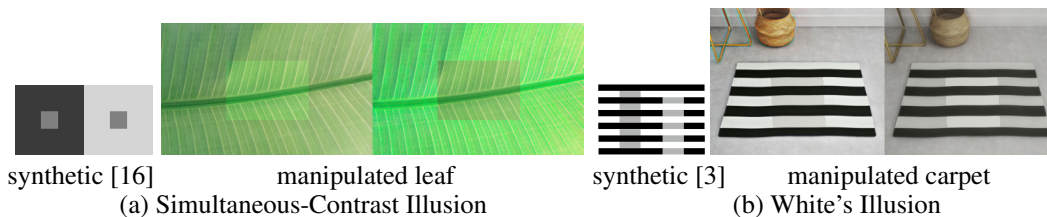

| synthetic [16] | manipulated leaf | synthetic [3] | manipulated carpet |
| :-: | :-: | :-: | :-: |
| (a) Simultaneous-Contrast Illusion | | (b) White's Illusion | |

Figure 1: **Examples of visual illusions.** (a) Look at the central rectangles. Do they have the same intensity/color? It seems they do not, but they actually do. (b) The same effect happens for the gray long rectangles—they are perceived as having different intensities, although their intensities are equal. This paper provides a general, data-driven explanation to these (and other) illusions. It also presents a model that generates illusions by modifying given natural images in accordance with the above explanation (the right illusions in (a) & (b)). Please watch the illusions on a computer screen.

## 1 Introduction

The physical state of the world differs from our subjective measure of its properties, such as lightness, color, and size, as demonstrated in Fig. 1. This gap leads to visual illusions, which is a fascinating phenomenon that plays a major role in the study of vision and cognition [7]. This, in turn, has made human-made illusions very popular; they are used for education, in business, and as a form of (op-)art and of entertainment (e.g. "Illusion of the Year Contest") [39, 50, 51].

A variety of explanations for visual illusions have been proposed in Psychology and Neuro-biology. The top-down approach argues that past experience is involved in visual processing, therefore affecting perception (and visual illusions) [20, 21, 29, 30]. Conversely, bottom-up models propose explanations for common visual illusions that do not rely on cognition or on prior knowledge [5, 6, 15, 31, 32, 40, 42]. Instead, they focus on biological mechanisms present in the human visual cortex. Input-driven changes of visual sensitivity, however, may occur in such models over time or evolution [14, 36].

Leaving the argument between these approaches aside, in both approaches retinal image statistics may play a major role (through life experience or evolution), which ask for the study of visual input.

This paper is inspired by the *wholly-empirical* evolutionary paradigm [33, 34, 36, 38], whose origins go back a hundred years [49]. In the context of illusions, it is motivated by the observation that a 2D projection of a 3D world makes the inverse optical problem ill-posed. For instance, the measured luminance of a surface corresponds to infinite possible combinations of environmental illumination, surface reflectance and atmospheric transmittance. Visual perception does not aim to recover real world properties (e.g., color, length and orientation) explicitly, as it is impossible. Instead, vision generates useful perception for successful behaviors without recovering real-world properties.

Evolution and lifetime learning that promote successful behavior also determine our interpretation of the scene. Thus, the statistics of the projections of natural scenes and the likelihood of patches in them determine our perception. Visual illusions are caused when the ill-posed inverse problem is interpreted statistically and this interpretation contradicts the original stimuli. This interpretation is correlated with the frequency of occurrence of scale-invariant retinal image patterns.

For instance, in Fig. 1(a) two identical gray (/green) boxes are surrounded by different backgrounds. They are perceived differently: The one on the bright background looks darker than the other. Fig. 1(b) demonstrated the opposite effect, as identical gray blocks look lighter when surrounded by an overall brighter background. How would one explain it? It turns out that natural patch statistics can explain both illusions and many more [22, 33, 34, 37, 44, 53, 54].

The *wholly-empirical paradigm* was supported by psycho-physical studies with limited data. The era of deep learning and big data provides the means to support the theory based on solid ground. This is a major goal of this paper: provide a general and unified explanation to seemingly-unrelated visual illusions. This goal is inline with the classical motivation of computer vision algorithms—mimicking cognitional mechanisms, some of which might be affected from this reality-perception gap.

Towards this end, we introduce a statistical tool that estimates the likelihood of image patches to occur, based on the learned statistics of natural scene patches. We will show how this tool assists us to explain three different types of well-known visual illusions.

An important property of our proposed tool is being reversible. This enables a controlled statistical manipulation of image patches, i.e. making a patch more (or less) likely with respect to a natural dataset. Thus, it allows us not only to explain visual illusions, but also to create ones, as illustrated in Fig. 1. Unlike the synthetic illusions that can be found in textbooks, our generated illusions appear as natural images, as after all, these are the illusions of our everyday life.

We are not the first in computer vision to be fascinated by visual illusions. In 2007, Corney et al. [10] proposed to use a shallow neural network and predict surface reflectance in synthetic images. This work managed to show that this network was deceived by several lightness illusions similarly to humans. A decade later, Gomez-Villa et al. [17, 18] trained deep convolutional neural networks (CNNs) on datasets of natural images. The networks were trained to perform low-level vision tasks of denoising, deblurring and color constancy. However, it was demonstrated that each network (one trained for denoising, one for deblurring and one for color constancy) is deceived by some illusions, but not by all. One may view these works as implicit support to the connection between the statistics of natural images and visual illusions. In [52] a GAN was trained to generate illusions out of a dataset of visual illusions. It was claimed that this approach is unable to fool human vision. In [19] synthetic visual illusions were generated, by adding an illusion discriminator that quantifies the perceptual difference between two target regions [18], to a GAN that generates backgrounds for the targets. The choice of the pre-trained illusion discriminator and the balance of the losses of the discriminators lead to different kinds of results, thus lacking generality.

We note that similar ideas to [34, 38] were suggested. Specifically, in [4] it was proposed that matching the sensors to the statistics of the stimuli in order to reduce redundancy in the response could lead to visual illusions. More generally, recent uniformization techniques such as *Sequential Principal Curves Analysis (SPCA)* have been proposed to explain the emergence of illusions when environment is changed [26]. Nonlinear transforms for error minimization [28, 45] may also be achieved by SPCA, thus giving alternative statistical explanation for illusions.

This paper makes three contributions: (1) It introduces a novel generative statistical tool for estimating the likelihood of image patches (Section 2). (2) It provided explicit support to the connection between

patch statistics in natural scenes and visual illusions, using a large dataset of natural images. It demonstrates that a single unified method can explain a variety of illusions (Section 3). (3) To complement the explanation of illusions, the paper proposes a general method to automatically generate "natural" visual illusions in order to demonstrate the effects in natural contexts. This is done by manipulating the likelihood of image patches (Section 4).

## 2 Measuring Patch Likelihood

To support input-driven approaches, statistics of patches in the real world must be provided. In the pre-big data era, this was difficult to do. Thus, the analysis was evaluated on small datasets (up to 4000 images) [48]. This section introduces a model to measure patch likelihood in large scale.

While patch likelihood is not measured explicitly in computer vision tasks, implicitly, it has tremendous importance in image restoration [11, 43, 56]. We seek after a general explicit method, which is not application-dependent. We require that this method would have generative capabilities, in order for it not only to assist to explain visual illusions, but also to generate ones.

Section 2.1 proposes a patch likelihood estimation model (Fig. 2) that can efficiently and accurately learn a high-dimensional distribution of a large dataset of natural scene patches. This raises an interesting question of how to evaluate the behavior of the proposed model. In the patch case, visual assessment of sampled patches is irrelevant and a quantitative ground truth does not exist. In Section 2.2 we introduce two measures, one is quantitative and the other is qualitative.

### 2.1 Framework

Since we aim at explicitly estimating the likelihood of properties (e.g., intensity, saturation etc.), as well as modifying these properties, we turn to likelihood-based generative models. These models can be classified into three main categories: (1) *Autoregressive models* [46, 47] (2) *Variational Autoencoders (VAEs)* [8, 23] and (3) *Flow-based models* [12, 13, 24]. We focus on the flow-based model, for three reasons: First, it optimizes the exact log-likelihood of the data and enables log-likelihood evaluation. Second, the model learns to fit a probabilistic latent variable model to represent the input data, which is important for the specific generative property we seek after. Third, the model is reversible.

We base our framework on *Glow* [24], a recent flow-based architecture. Hereafter, we briefly introduce the theory behind this model, adapted to our patch case (as [24] handles full images). Let $x$ be a patch, sampled from an unknown distribution of natural scene patches $p^*(x)$. Let $D$ be a dataset of samples $\{x_i | i = 1, ..., N\}$ taken from the same distribution. We seek a model $p_\theta(x)$ that will minimize

$$\mathcal{L}(D) = \frac{1}{N} \sum_{i=1}^{N} -\log(p_\theta(x_i)). \tag{1}$$

The generative process is defined by the latent variable $z \sim p_\theta(z)$, where $p_\theta(z)$ is a multivariate Gaussian distribution $N(0, I)$. The transformation of the latent variable to the input space is done by an invertible function $x = g_\theta(z)$ s.t.

$$z = g_\theta^{-1}(x) = f_\theta(x). \tag{2}$$

The function $f_\theta(x)$ is a composition of $K$ transformations, termed a *flow*. We denote the output of each inner transformation $f_i$ with $h_i$. Then, the probability density function of the model is:

$$\log(p_\theta(x)) = \log(p_\theta(z)) + \sum_{i=1}^{K} \log(|\det(dh_i/dh_{i-1}|). \tag{3}$$

For the families of functions used in flow-based architectures, this term it efficient to compute. Furthermore, both the forward path and the backward path are feasible.

**Implementation.** Fig. 2 illustrates our model. It is based on the architecture of Glow, with a single *flow* and $K = 32$ composed transformations. The input consists of image patches of size $16 \times 16$, a size that manages to capture textured structures and to allow stable training. The network was trained on random patches, sampled from *Places* [55], which is a large scene dataset. As such, multi-scale patterns are inherent in this large and rich dataset, even when sampling patches of a single size.

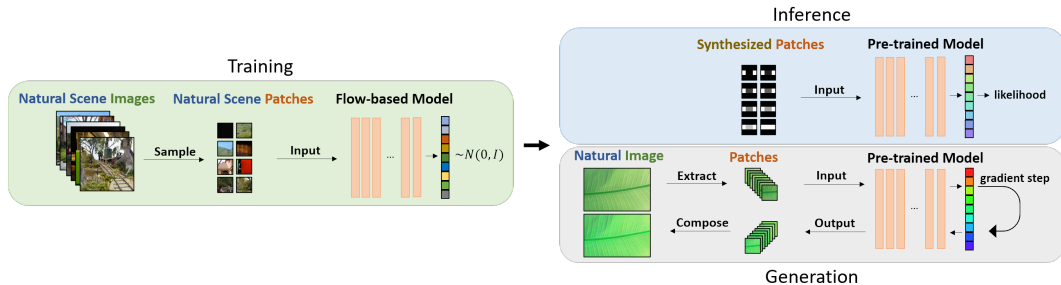

Figure 2: **Framework.** During training, the likelihood of natural scene patches is learned. During inference (Section 3), this pre-trained model is fed with synthetic patches that represent common visual illusions, and estimates their likelihood. To generate visual illusions given an image (Section 4), the image patches are first extracted and their corresponding latent variables are manipulated with a gradient step. They are then fed back into the reverse path, in order to reconstruct a new image.

## 2.2 Evaluation

There is neither ground truth for patch likelihood, nor commonly-used evaluation; thus we propose two evaluation measures for our model.

*Quantitative evaluation—Center of patch test.* Many of the experiments of [34] were based on uniform sampling of natural scene image patches and calculating the probability of hand-crafted features, e.g. the color of the center patch given its surroundings. This approach requires sampling of many patch templates, for instance different colors of the center area or the surrounding.

We propose a similar, yet simpler & more general approach: We generate the target patches (e.g., different values of center & surrounding) and inject them to our pre-trained network. It provides us with a likelihood score for each patch, which can be compared to [34]. Section 3 demonstrates that indeed the results are amazingly similar for the simultaneous-contrast illusion they studied.

*Qualitative evaluation—Min-Max patch test.* Sorting patches of an image by their network's scores may also provide a sanity check for the network's behavior. We would expect smooth patches to be more likely than textured ones, and among the textured patches we would expect a reasonable ranking—one that expresses the learned statistics. Since it is impossible to determine whether a ranking is reasonable with respect to a large dataset of images, we propose to determine it by training on patches of a single image. This would allow visual evaluation of the results. Furthermore, a comparison of the ranking of the internal statistics (relative to single image) to that of the external statistics (relative to the entire dataset) could help in the evaluation of our tool.

Fig. 3 demonstrates our results on two images. For each image, we present 8 random patches from the internal most/least likely 100 patches and 8 patches from the external most/least likely 100 patches (trained on Places). The most likely internal and external patches are very similar—they are both very common in the source image and are relatively smooth. There is, however, a clear difference between the least likely patches: The internal least likely patches are indeed very unique in the source images (the white windows and the red-on-orange pattern). This confirms our sanity check! We may now believe our tool, according to which the external least likely patches are the green-brown-white and red strings on blue background, respectively. These patches are unique in the world, even though they appear much more in the source images. (Recall that our model uses only external patches.)

## 3 Explaining Visual Illusions

Equipped with our deep learning-based probabilistic tool, this section attempts to support the input-driven empirical paradigm regarding deceived perception of common visual illusions,. The underlying assumption is that the statistics of features in large datasets of natural scenes (Places [55]) is similar to that in the "dataset" of retinal images. We introduce a general technique for explaining visual illusions, given patch statistics. Furthermore, we demonstrate the statistical reasoning on three visual

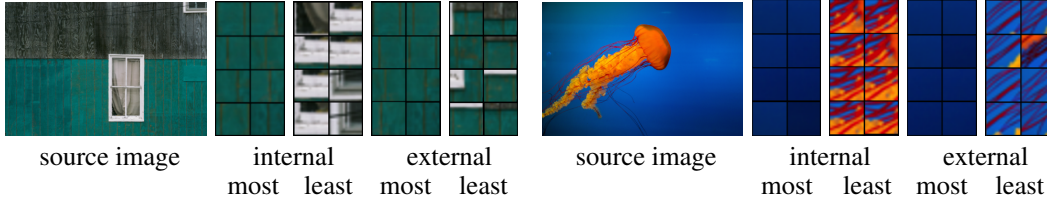

| source image | internal | | external | | source image | internal | | external | |
| | most | least | most | least | | most | least | most | least |

Figure 3: **Min-Max patches.** Samples from 100 most-likely and 100 least-likely patches of images (from [1]), w.r.t. the internal image statistics (trained on a single image) & the external statistics (trained on Places). While the most-likely patches are smooth in both cases, the least-likely patches differ. For instance, white patches (left) are very likely in the world, but are unlikely in the image.

illusions, focusing on intensity & color illusions. For each illusion we ask what the likelihood is of the illusory pattern to appear in a projection of a natural scene and use this likelihood for ranking.

We use the common *percentile rank*, defined as the percentage of scores that are smaller or equal to it. It is the *Cumulative Distribution Function (CDF)*, normalized to the range of $[0..100]$. The percentile rank is empirically shown to correspond to the perceived intensity or color [35, 38]. The reason is that the relative number of times biologically-generated patterns are transduced and processed in accumulated experience tracks reproductive success. Thus for instance, the frequencies of occurrence of light patterns over time is "aligned" with perceptions of light and dark.

**Method.** Given an illusion, we define a template & target for it. For instance, in the case of Fig. 1(a), the template is a rectangular surrounding and the target is the inner rectangle. Then, we generate 256 instances of the template with the same surrounding (context) but with different values (of intensity, saturation, hue or value) of the target area. This yields 256 patches, which differ from one another only in the target area. Our goal is to evaluate the likelihood of these patches, as it expresses the probability of the target values given a specific context. This is done by providing the pre-trained network from Section 2 with patches as input. The system returns the likelihood of this pattern.

Let $A, B$ be two different backgrounds of the same target area, having value $T$. If the percentile rank of $T$ in background $A$ is higher than that of value $T$ in background $B$, this means that statistically we expect the target area to have a higher value (e.g. lighter, in the case of intensity) in $A$ than in $B$. Therefore, the perceived value in the target will be higher in $A$ than in $B$. In terms of the likelihood function, this means that the peak of the likelihood value in $A$ is attained in a lower value than in $B$.

**Illusions.** Hereafter we demonstrate the results of our method on three different types of illusions. We note that a statistical explanation of the first illusion was given by [34], whereas similar statistical explanations of the other two illusions have not been discussed in the literature.

*1. Simultaneous lightness/color contrast illusion [16, 27].* In this illusion, two identical patches are placed in the center of different backgrounds. While the color of these central patches is the same, it appears darker when surrounded by a brighter background than by a darker background (Fig. 4(a)).

Our experiment is performed both on the hue, saturation and value (in HSV color space) and on the intensity (in gray-scale). The template is a uniform background and a uniform center area (target). For each property (e.g. saturation), we set a value in the range $[0, 255]$ and generate 256 patches in which the background has this value; the value of the center of the patch increases from 0 to 255.

Fig. 4(b) shows the saturation likelihood graphs for (a), as outputted by our network. The likelihood is maximal when the center and the surrounding have the same saturation and drops as they get farther away from each other. Fig. 4(c) shows that the same saturation $T$ of the center areas has a higher percentile rank when surrounded by less saturated (lighter) background than by more saturated background; hence, it is perceived as darker. The same analysis applies to the hue & value of the HSV color space. This result is very similar to the empirical experiments of [34] for this illusion.

This example demonstrates a key strength of our model—being capable of generalizing well from natural patches it is trained on to synthetic patches it is fed with in the analysis. As an alternative, we performed a couple of experiments with GMMs (with up to 200 components). The generalization was inferior, i.e. the likelihood graph of the simultaneous-contrast illusions (corresponding to Fig. 4) is almost a delta function.

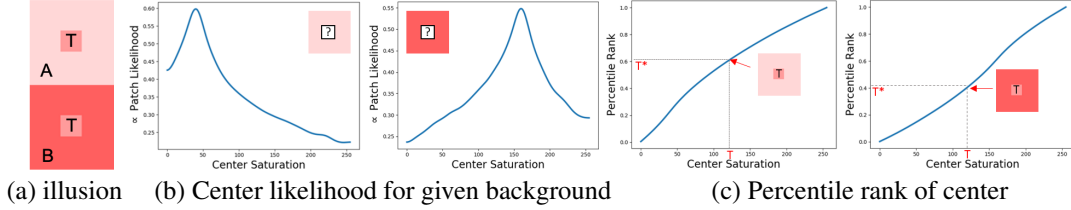

(a) illusion     (b) Center likelihood for given background     (c) Percentile rank of center

Figure 4: **Experiment—contrast illusion** (a) The backgrounds have the same hue & value and different saturation. The center, of color $T$, is perceived more saturated on the top than on the bottom, though it is the same. (b) shows the likelihood of the saturation value of the center when surrounded by the top/bottom background; the peaks are when the value $T$ is that of the background. (c) Since the percentile rank of $T$ on top is higher than on the bottom, it will be perceived as more saturated.

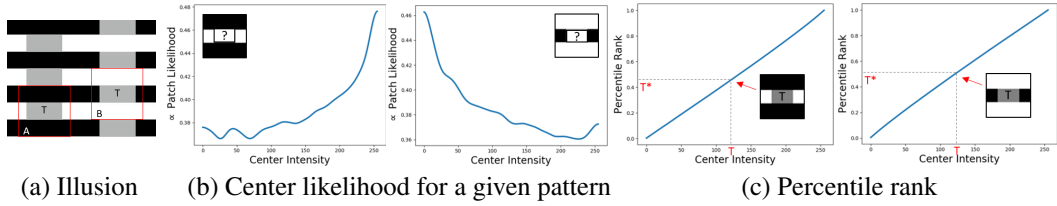

(a) Illusion     (b) Center likelihood for a given pattern     (c) Percentile rank

Figure 5: **Experiment—White's illusion** (a) The targets $T$ have the same value, but they are perceived differently, depending on their surrounding patterns $A, B$ (in red). (b) The likelihood of the target grayscale intensity, when interrupting the white(/black) bar. (c) When the target area interrupts the white bar, it has a lower percentile rank, and hence it looks darker, and vise versa. The same statistical paradigm explains both White's illusion and the simultaneous-contrast illusion, although they cause an opposite contrast effect.

*2. White's illusion.* The black and the white horizontal bars are interrupted by identical target gray blocks (Fig. 5(a)). These blocks appear darker when they interrupt the white bars and lighter when they interrupt the black [3]. Interestingly, the illusory effects of the edges of the target gray blocks in White's illusion and in the simultaneous-contrast illusion, are reversed. Here, when the target block has more of dark edges it looks darker and when it has more of bright edges it looks lighter.

In our experiment, in the first template, the pixels of the top & bottom thirds are black and the middle bar is split horizontally, such that the left and the right quarters are white. The target area interrupts the middle bar and its value increases from $0$ to $255$, leading to $256$ patches. In the second template, the roles of the black and the white are reversed.

Fig. 5(b) presents our results–the likelihood graphs of the target value, given its surrounding. These graphs support the findings: When the gray block interrupts the white bar, it is more likely to be light, thus it has a lower percentile rank, and vise versa (Fig. 5(c)). As before, this low percentile rank explains the dark appearance when interrupting the white bar, while the high rank explains the light appearance when interrupting the black bar.

*3. Hermann grid.* Fig. 6(a), which consists of uniformly-spaced vertical and horizontal white bars on a black background, illustrates this illusion. Stare at an intersection; this intersection appears white when it is in the center of gaze, but gray blobs appear in the peripheral intersections [41]. This illusion relates not only to the statistics of patches, but also to the receptive field, which is smaller in the center of gaze than in the periphery.

Therefore, to explain this illusion we must also emulate the receptive field. This is done by considering a low-scale image as effectively corresponding to a large receptive field, and a high-scale image as corresponding to a smaller receptive field. We feed our network first with patches of a high-scale grid ($512 \times 512$) and then with patches of a low-scale image ($256 \times 256$), but with the same patch size.

The difference in the likelihood maps of the two cases can be observed in Fig. 6(b)-(c), as heat-maps for each patch. In high-scale, the white intersections are highly likely (brown). This is not surprising, as it represents the small receptive field, which captures the center of the intersections as smooth and

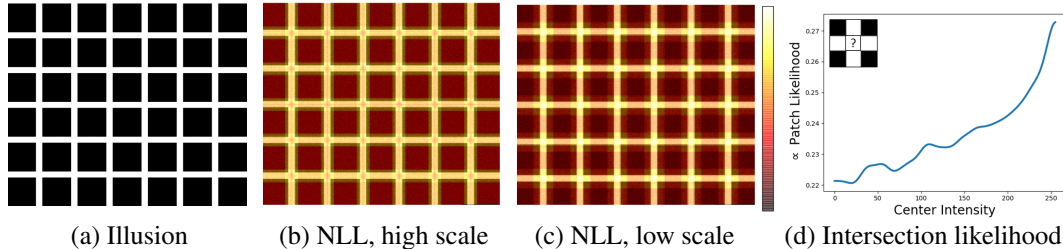

| (a) Illusion | (b) NLL, high scale | (c) NLL, low scale | (d) Intersection likelihood |

Figure 6: **Experiment—Hermann grid.** (a) Stare at one specific white intersection. It appears white when it is in the center of the gaze, but illusory gray blobs appear when not at the center. (b) In the patch *negative log-likelihood (NLL)* map of the grid, the white intersection is likely (brown) in high scale and (c) unlikely (yellow) in low scale. (d) presents the likelihood graph of the gray-scale value of the center of an intersection in low scale. Due to the low resolution in the peripheral vision, the value of the center has a lower percentile rank and would therefore be perceived as darker.

likely white patches. However, in low scale, the white intersections become unlikely (yellow). This is so because white crosses on a black background are indeed unlikely in real life.

To explain why the peripheral intersections look darker, we employ our method. The template in this case is a white cross on a black background, which looks like the intersection area in low scale. The target area is the center of the cross and its value increases from $0$ to $255$, leading to 256 different patches. Fig. 6(d) shows the results: the likelihood of the value of the center of the cross increases as the gray-scale value approaches white. In the periphery, where these intersections are not pure white [2], they would have a lower percentile rank and therefore would look darker, as before.

## 4  Generating Visual Illusions

This section introduces a novel method for generating visual illusions, which is considered to be a grand challenge [19]. We aim at generating illusions in the context of natural images, by enhancing illusory effects in a given image (Fig. 7). This requirement adds two new difficulties, in comparison to generating synthetic illusions. First, it is impossible to choose a uniform target area. Second, due to the amount of details in a natural image, the target area should be large in order to be noticeable. However, if the target is large, since the neighborhood of its inner parts do not change, the illusory effects might be reduced (since they depend on the surrounding, including adjacent inner parts).

The key idea of our approach is based on the principle of the empirical paradigm: We generate illusory effects by controlling the likelihood of image patches. In particular, given an image, we generate context (surrounding) that is slightly more likely or slightly less likely, as described hereafter.

**Method.** Given an image and target areas, the algorithm first extracts all the image's overlapping patches, except for those of the target. Second, these patches are fed-forward into our pre-trained network (Section 2), resulting with a latent variable $z$ (Eq. 2) and a likelihood score for each patch. Third, a gradient step is performed on the latent variable, such that the associated patch's likelihood would slightly increase or slightly decrease. A manipulated patch is generated by injecting the above manipulated latent variable into the reverse pass of the network, which generates its corresponding manipulated patch. Finally, the manipulated patches compose (by averaging corresponding pixels) a similar image to the input image, except that each patch (excluding the target) has a new likelihood.

The core of this method is the controlled likelihood manipulation of image patches. This operation is feasible due to (1) the reversibility of our network and (2) the form of the latent variable. We modify $z$, while our goal is to modify the likelihood of its corresponding patch $x$. This will indeed happen, since as observed by [12], regions with high density, in patch (input) space, shall also have a large log-determinant value and a large value of $p_\theta(z)$ (Eq. 3).

We use our prior knowledge regarding the distribution of the latent variable to manipulate input patches according to their likelihood. Let $x$ be a patch and $z = f_\theta(x)$ be its latent representation. Manipulating the latent variable $z$ to $z'$ and back-projecting it with the reversible flow to the patch

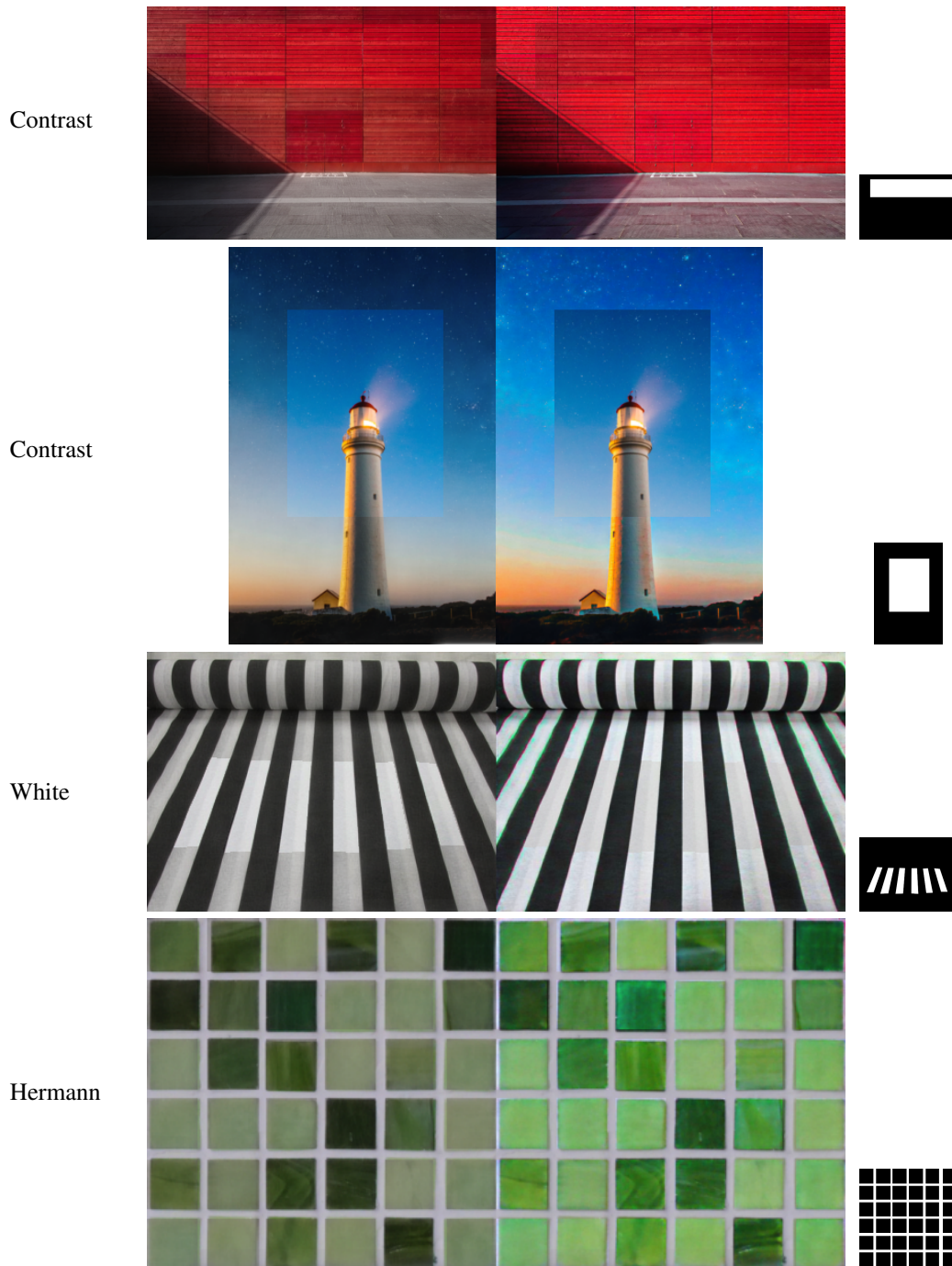

(a) Illusion                  (b) Result                  (c) Mask

Figure 7: **Generated visual illusions.** Each input image was manipulated patch-wise, changing the surrounding of the masked area (shown in white on black). (a-c) While the masked regions have the same color in the left and in the right images, they are not perceived as so. (d) The illusory gray blobs in the white intersections are enhanced in the left image and are reduced in the right. More examples are given in the supplemental material. Please watch this figure on a computer screen.

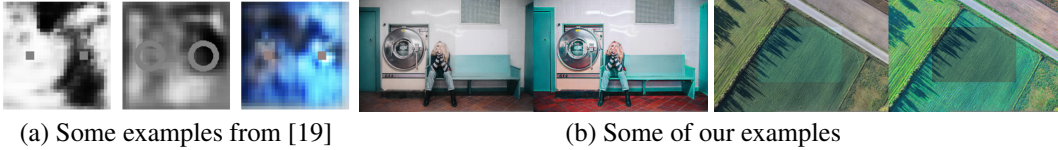

(a) Some examples from [19]　　　　　　　　(b) Some of our examples

Figure 8: **Comparison of simultaneous-contrast illusions.** (a) In [19] the background of identical target areas (rectangles or rings) is automatically generated, after training a GAN with an illusion discriminator on DTD [9] or CIFAR10 [25] datasets. (b) Instead of generating backgrounds from scratch, we manipulates the context (backgrounds) of any target area in a given natural image.

space result in a patch $x' = f_\theta^{-1}(z')$. Applying manipulation operation $\Psi$ yields:

$$x' = f_\theta^{-1}(z') = f_\theta^{-1}\Big(\Psi\big(f_\theta(x)\big)\Big). \tag{4}$$

As the distribution of $z$ is Gaussian (Section 2), $\Psi$ is implemented as a simple gradient step:

$$z' = \Psi(z) = z + \eta \cdot \Big[ -z \cdot e^{-\frac{z^2}{2}} \Big]. \tag{5}$$

The step size $\eta$ determines the amount of likelihood manipulation. In our experiments, $\eta_1 = 0.6$ to increase the likelihood and $\eta_2 = -0.8$ to decrease it.

**Results.**  In our results, the manipulation is not limited to a single property of the image, (i.e. saturation). Instead, patch modification based on the likelihood results in changing different properties of the image for each patch, depending on the input itself. While the art in Op-Art is to produce these effects manually, by editing the image properties, our approach achieves these effects automatically.

Fig. 7(a-b) shows simultaneous-contrast illusions. Two identical target areas (the white area in the mask) are perceived as having different colors, thanks to their different backgrounds. The left background was generated by manipulating the source image with $\eta_1$, and the right with $\eta_2$. Fig. 8 compares our results to that of [19], where illusions of this type were synthesized by adding a pre-trained block that acts as an illusion discriminator in a GAN framework.

Fig. 7(c) illustrates a White-like illusion. Again, the left background was generated by manipulating the source image with $\eta_1$ and the right with $\eta_2$. The targets that interrupt the fabric's darker stripes (left) look lighter, although surrounded from right and left by brighter stripes than in the right image.

Fig. 7(d) demonstrates a Hermann grid-like illusion, as generated by our system. In these two natural grids the white lines are the same, but the colored squares were manipulated as before (left with $\eta_1$, right with $\eta_2$). The gray illusory blobs are enhanced in the left image, where the blocks were manipulated to be more likely, and reduced in the right. More results are given in the supplementals.

## 5   Conclusions

The empirical paradigm of vision argues that illusions result from the attempt to statistically interpret an ill-posed inverse problem. In this paper we support this paradigm by proposing a unified method, which is able to explain a variety of lightness and color visual illusions, by analyzing the statistics of image patches in big data. Furthermore, the paper shows that reversing the process, by changing the likelihood of patches in an image, manages to enhance visual effects for the same analyzed illusions. Both the support of the paradigm and the generation of illusions are possible thanks to a novel tool that measures the likelihood of image patches and has generative capabilities.

There are several interesting future directions. First, we aim to automatically choose the best target areas for the illusion generation process. Additionally, developing quantitative evaluation metrics for generated illusions is a challenging direction [19]. Finally, more illusions could be studied using the proposed tool, both color-based and geometric. Empirical explanations exist for some additional illusions, such as mis-perceiving size and direction.

---

Our code is available at `https://github.com/eladhi/VI-Glow`

## Acknowledgements

We gratefully acknowledge the support of the Israel Science Foundation (ISF) 1083/18.

## Broader Impact

*a) Who may benefit from this research?* This work promotes knowledge regarding visual illusions. Rather than focusing on biological circuitry, this work proposes a tool for studying statistical properties shared by a variety of visual illusions. The patch-likelihood tool may interest researchers & developers in computer vision and machine learning. The illusion analysis and illusion generation methods may interest not only academic researchers (in psychology, neuro-science, computer vision, and machine learning), but also other groups that use visual illusions for educational purposes, in entertainment, in advertisements, in business programs, and in (op-)art.

*b) Who may be put at disadvantage from this research?* The analysis and the results of this paper cannot serve in a negative manner. However, generating visual illusions might be used to deceive people, for instance in advertisement: An object might look of a certain color (or size), while its true color differs.

*c) What are the consequences of failure of the system?* We do not see any such consequences.

*d) Does the task/method leverages biases in the data?* This is not applicable.

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
