[Supplementary Material 1 · Implementation_Details.pdf]

# Supplementary Material - Implementation Details

The architecture is based on the work of Glow. Our adaptations are as follows:

- Input Size: $16 \times 16 \times 3$ (RGB patches sampled from Places dataset)
- Batch Size: 256
- L (number of flows): 1
- K (number of transformations in a flow): 32
- Channels: 512
- Optimizer: Adam
    - Learning Rate: 1e-4
    - Betas: (0.9, 0.999)
- No use of LU decomposition

During training, we randomly sample a batch of images from the dataset and first resize the images such that the smaller edge of the image is of size 256 pixels. Then, we randomly crop each image to a single $16 \times 16$ patch, leading to the same batch size of patches. We also apply random horizontal flip.

For image manipulation, we extract all overlapping patches (sliding window) of the source image and input them in batches to the forward pass of the network. After performing the gradient step on the latent variable, we input it (same batch size) to the reverse pass of the network. Then, the image is composed such that every pixel is the mean of its values in all the patches it appears in.

Our code is available at `https://github.com/Eladhi/VI-Glow.`



[Supplementary Material 2]

# Supplementary Material - Center of Patch Test

The center of patch test helps us to evaluate the training process, as described in Section 2. Here we provide examples of graphs that show the expected behavior - the maximum likelihood is attained when the center of the patch and its surround are the same. We show it in gray-scale intensity, hue, saturation and value (HSV color space).

Each graph shows the likelihood of the patch, for a given surround, vs. the value of the center. The rows represent the property (intensity, hue, saturation or value) that is being changed for the center area, and the columns represent the corresponding value of the surround. For example, in (1,1) the surround has a gray-scale value of $60/255$ and the gray-scale value of the center varies from 0 to 255, while in (2,2) the surround has a hue of $120/255$ and the hue of the center varies from 0 to 255. Moreover, note that the range of hue values is cyclic.

Figure 1: **Likelihood of patches for a given surround vs. value of the patch center.**



[Supplementary Material 3]

# Supplementary Material - Min-Max Patch Test

We provide more examples of the min-max patch test, used in Section 2. The patches of each image were sorted by their likelihood scores:

1. When our model was trained on a single image (the source image; internal statistics)
2. When our model was trained on Places dataset (external statistics)

This experiment is used for a qualitative evaluation of the training process. We present a sample of the top/least 100 likely patches w.r.t. the internal/external statistics.

| source image | internal most likely | internal least likely | external most likely | external least likely |

Figure 1: **Min-Max patches.** Samples from 100 most-likely and 100 least-likely patches of images, w.r.t. the internal image statistics (trained on a single image) & the external statistics (trained on Places dataset).



[Supplementary Material 4]

# Supplementary Material - Additional Results

Additional examples of contrast-like illusions:

(a)

(b)

(c)

(d)

(e)

(f)

(g)

(h)

(i)

(j)

Figure 1: **Contrast-like visual illusions.** (a-l) The masked (white) area is the same in the left and right images.

Additional examples of White-like illusions:

Figure 2: **White's-like Visual illusions.** The masked (white) area is the same in the left and right images.

Additional examples of Hermann-like illusions:

Figure 3: **Hermann-like Visual illusions.** (enhanced in the left, reduced in the right) The masked (white) area is the same in the left and right images.