[Reviews · NeurIPS 2020]

Review 1

Summary and Contributions: This paper presents a model to explain visual illusions based on considering patch likelihood. This presents a novel take on the impetus for a certain set of illusions and is an interesting casting of the problem.

Strengths: The model appears to be successful in reproducing several phenomena that give rise to visual illusions based on patch likelihood. The images corresponding to visual illusions that are generated are interesting. The significance and novelty of the paper seem relatively strong although I am not an expert in this area. This is clearly relevant to the NeurIPS community.

Weaknesses: The assumption that patch likelihood is appropriately measured could use some more justification. Additionally, there could be more examples of similar phenomena explained by the model.

Correctness: Claims and methods in the paper appear to be correct. Empirical methodology also appears to be well done.

Clarity: The paper is clear and well written.

Relation to Prior Work: To the best of my knowledge, the coverage of previous contributions is adequate.

Reproducibility: Yes

Additional Feedback: In light of all the additional context, rebuttal and discussion I am inclined to stand by my original score for this paper.


Review 2

Summary and Contributions: # POST REBUTTAL I have decided to keep my score as it is - the authors have offered clarifications for the concerns and questions I had. I think is very interesting work and hope to see it get in the conference. # This paper presents a framework for explaining (some) visual illusions using patch statistics. A patch generative model is learned which can estimate the likelihood of a given patch. It is shown that under several controlled settings the percentile rank of the patch is a good indicator to which interpretation the ROI in the patch would be perceived. Furthermore it is shown that by modifying the likelihood of patches in a masked area in an image and regenerating them it is possible to create some simple illusions in a natural image.

Strengths: I think this is a very nice paper which has several strengths: * The paper and general approach are interesting * The method is demonstrated to work well on three illusions which share some inherent properties, but are different enough to make a convincing case. * It is shown that to some extent an illusory image can be generated by modifying the likelihood of patches - this requires a bit of careful masking in my eyes, but still neat. * An actually interesting and relevant use for generative models which goes beyond generating nice images of faces or dogs.

Weaknesses: There are several issues here which I would like the authors to address: * Could the authors comment on the use percentile rank? I understand the reasoning behind it more or less but this is not explained in the paper at all. * What is the relationship between the CDF and percentile rank in this case? is there a way to express one with the other? * The experiments show that in a controlled setting (where a clear target patch and template patch are defined) it is possible to explain several illusions. One thing which is common to all the illusions is that the target patch is flat - what about cases where the patch to explain may have some structure? like the Kanitze triangle? this would make a much more convincing case for the method. * The authors show that the percentile rank correlates with the perceived *relative* lightness (for example) but they do not show if this is actually at the same scale of perception - do subjects report the same change in lightness perception? (I'm sure these numbers can be found in literature). * Only one generative model is tested here - do results change with other models? say a simple GMM or a sparse coding based one?

Correctness: Seems so.

Clarity: The paper is nicely presented and clearly written.

Relation to Prior Work: Seems good.

Reproducibility: No

Additional Feedback: More information about the actual model implementation and networks used would be useful.


Review 3

Summary and Contributions: The authors use current generative models based on invertible normalizing flows to propose an interesting update of classical statistical ideas from the visual neuroscience community to explain visual illusions occurring in biological vision systems. -- Post Rebuttal --: I've kept my positive score and am happy with the rebuttal sent by the authors addressing some of my concerns. I encourage the authors to include the suggested citations

Strengths: The strength of the paper is the proposition of a unifed approach for explaining visual illusions and generating them. Visual illusions are interpreted according to the departure of images from the expected behavior. Relation between a physical magnitude (e.g. luminance or chromatic purity) and the corresponding perceptual magnitude (lightness or saturation) is given by the cumulative density function of natural images along the physical dimension. The likelihood of image patches is computed from an invertible flow presented at NeurIPS-2018. The invertibility of the flow allows to synthesize new images with the desired likelihood, leading to changes in the perception. The use of this recent generative model makes conclusions solid and allows systematic evaluation of the claims. The proposed explanation of illusions may clarify the statistical principles that shape the behavior of the visual brain. Unveiling these principles is a fundamental goal of the Neural Information Processing community.

Weaknesses: The only weakness of the work is on the relation with previous literature: statistical explanations of illusions were suggested before Purves et al., and alternative uniformization and Gaussianization techniques have been proposed to implement these ideas. See the specific connections below in the "relation to prior work" question.

Correctness: There is no technical problem in the proposed methodology. Actually, the use of normalizing flows to explore the relation between the biological behavior and the statistics of visual input is practical and promising.

Clarity: Methodology is properly explained and the description allows the reader to train the flow and compute the corresponding probabilities. The synthesis of new images from the inverse of the flow should stress that one should select solutions along the considered physical dimension (e.g. luminance or chromatic purity). This constraint should be more crearly stated because if not (as far as I understood) a target probability may be obtained by very different images/contexts.

Relation to Prior Work: On the one hand, the current text is too focused on the contributions of Purves et al. Fantastic papers of Purves et al. are very inspiring, but similar ideas were suggested before and this has to be acknowledged. Specifically, Horace Barlow proposed that matching the sensors to the statistics of the stimuli in order to reduce redundancy in the response (using a sort of linear ICA) could lead to visual illusions [Barlow90]. More generally, redundancy reduction or information maximization is connected to (nonlinear) Gaussianization and uniformization transforms. Therefore, more recent uniformization techniques such as Sequential Principal Curves Analysis (SPCA) have been proposed to explain the emergence of illusions when environment is changed [Lapàrra15]. Nonlinear transforms for error minimization [Twer01,McLeod03] may also be achieved by SPCA, thus giving alternative statistical explanation for illusions [Laparra15]. Moreover, error minimization also explains similitudes between visual illusions in artificial neural networks and human viewers [Gomez-Villa20]. On the other hand, regarding invertible flows, the selected Glow transform is very similar to previous invertible Gaussianization transforms such as [Laparra11]. That rotation-based Gaussianization also identifies Gaussian latent spaces, it is able to compute the likelihood of individual patches and it is invertible so that the proposed methodology could also be implemented with [Laparra11]. [Barlow90] Barlow, H. (1990). “A theory about the functional role and synaptic mechanism of visual aftereffects,” in Vision: Coding and Efficiency, ed C. B. Blakemore (Cambridge, UK: Cambridge University Press), 363–375 [Laparra15] Laparra V and Malo J (2015) Visual aftereffects and sensory nonlinearities from a single statistical framework. Front. Hum. Neurosci. 9:557. doi: 10.3389/fnhum.2015.00557 [Twer01] Twer, T., and MacLeod, D. A. (2001). Optimal nonlinear codes for the perception of natural colours. Network 12, 395–407. doi: 10.1080/net.12.3.395.407 [McLeod03] MacLeod, D. A. (2003). “Colour discrimination, colour constancy, and natural scene statistics,” in Normal and Defective Colour Vision, eds J. Mollon, J. Pokorny, and K. Knoblauch (Oxford, UK: Oxford University Press), 189–218. [Gomez-Villa20] Gomez-Villa A, Martin A, Vazquez J, Bertalmio M, and Malo J. Color Illusions Also Deceive CNNs for Low-Level Vision Tasks: Analysis and Implications Accepted in Vision Research. https://arxiv.org/abs/1912.01643 [Laparra11] Laparra, V., Camps, G., and Malo, J. (2011). Iterative gaussianization: from ICA to random rotations. IEEE Trans. Neural Netw. 22, 537–549. doi: 10.1109/TNN.2011.2106511

Reproducibility: Yes

Additional Feedback: I think authors should include the suggested references to clarify (1) previous statistical explanations of visual illusions based on infomax and error minimization, and (2) the relation with previous invertible Gaussianization transforms.


Review 4

Summary and Contributions: The paper proposes a statistical theory and associated model to explain contrast-type visual illusions. In a nutshell, the authors train a flow model on the Places dataset, which can then be used to gauge the likelihood in the world of any 16x16 image patch from a test image. After the model is trained, a new illusion is segmented into a target region (which will be misperceived by humans) and a template region (which provides context that is believed to drive the misperception). The paper focuses on contrast-type illusions (over intensity, saturation, and hue), where a central patch's perception is influenced by its surroundings. Interestingly, the method is shown to be useful for generating new illusions.

Strengths: - interesting quantitative approach to analyzing illusions. - well written paper with engaging insights.

Weaknesses: - The authors propose to create a "single unified method can explain a variety of illusions" (line 70), but this is an overstatement. Only 3 types of illusions are studied (simultaneous contrast, White's, Hermann grid), which are actually quite similar within the overall very broad realm of visual illusions (which includes other dimensions like size constancy, collinearity/offset illusions, illusory contours/shapes, various 3D illusions like the Penrose stairs, dynamic illusions like the barber pole, breathing square, and many others, etc). - Thus, while the approach is interesting, its scope is quite narrow. It would help rather than hurt if the authors made that clearer (e.g., in the title, abstract, and contributions).

Correctness: The results are interesting and the approach appears correct, within its relatively narrow scope.

Clarity: The paper is well written but should clarify the breadth of scope better.

Relation to Prior Work: Good review of previous work.

Reproducibility: Yes

Additional Feedback: An interesting approach, yet it left this reviewer in search for a more general theory of visual illusions, which would reach beyond contrast-type illusions.

[Author Response · NeurIPS 2020]

We thank the reviews for their hard work, enlightening comments and positive feedback, appreciating the novelty and
the results: R1: "a novel take on the impetus for a certain set of illusions" R2: "a very nice paper... use for generative
models which goes beyond generating nice images of faces or dogs." R3: "Unveiling these principles is a fundamental
goal of the Neural Information Processing community." R4: " engaging insights".

Hereafter, we respond to the reviewers' individual comments.

**R1:** The assumption that patch likelihood is appropriately measured could use some more justification.
Since our model is a flow-based generative model, it optimizes the log-likelihood of the data (image patches, in this case)
during training. This, in turn, allows likelihood evaluation [22]. On the practical side, since there is no commonly-used
evaluation, nor a ground truth, for patch likelihood, we propose in Section 2.2 a couple of new evaluations: (1) a center
patch test that compares the likelihood of patches to the empirical results of [30] (quantitative) and (2) a min-max test
that compares the ranking of patches trained on a single image and on an external dataset (qualitative).

**R1:** There could be more examples of similar phenomena explained by the model.
Certainly. Our paper focuses on a variety of lightness/color illusions, which "share some inherent properties, but are
different enough to make a convincing case" (R2). However, the model may explain similar phenomena in geometric
illusions (our perception of lengths/angles as a function of their percentile rank) [20]. This is a major future direction.

**R2:** Could the authors comment on the use of percentile rank?...the relationship between the CDF and percentile rank
The percentile rank of a given value is the percentage of values in its frequency distribution that are equal or lower to it.
It is shown empirically in e.g., [31,34] (by analyzing the responses of human observers) that this relative percentile
ranks predict perception. The reason is that the relative number of times biologically-generated patterns are transduced
and processed in accumulated experience tracks reproductive success. Thus, for instance the frequencies of occurrence
of light patterns over time is "aligned" with perceptions of light and dark.
The percentile rank is the $CDF$, normalized to the range of $[0..100]$ (to be percentage).

**R2:** What about cases where the patch to explain may have some structure?
Thanks for the question! The White illusion is an example where the patch to explain has structure. In addition, there
are geometrical illusions (e.g. direction or length of lines) that can be explained by a similar theory of statistics of
natural images. Though this is beyond the current paper, it is an exciting future direction.
Kanizsa triangle: We are not aware of statistical explanations to this illusion. This is worth studying (in the context of
depth statistics). In the paper we provide novel statistical explanations to White & Hermann (a statistical explanation
for the simultaneous-contrast illusion has existed before).

**R2:** scale of perception - do subjects report the same change in lightness perception?
We have not found raw data for the specific illusions we study in the paper. For certain geometrical illusions, the
percentile rank is found to be at the same scale of perception, e.g. perception of line length [20]. We note that in order
to make conclusions regarding scale, the settings of the psychophysical experiments should be taken into account;
currently each experiment depends on specific parameters, such as the distance of the subjects from the monitor, the
size of the illusion itself and its inner structure.

**R2:** Do results change with other models... say a simple GMM...?
Indeed. The strength of our model is that it is capable of generalizing well from natural patches it is trained on to
synthetic patches it is fed with in the analysis (Section 3). We performed a couple of experiments with GMMs, which
exhibit inferior generalization. For instance, the likelihood graph of the simultaneous-contrast illusions is almost a
delta function. Another benefit of our model is being generative, thus it may be easily used for illusion generation
(Section 4); it is less clear how to do so with GMMs (simple sampling does not work well).

**R2:** More information about the actual model implementation and networks used would be useful.
The code will be released upon acceptance. Implementation details are provided in the supplementary materials; we
will add any requested information.

**R3:** The only weakness of the work is on the relation with previous literature.
Thank you very much for these references and the extra information. We will discuss the explanations of visual
illusions suggested in these papers, including the relation to the statistics of the stimuli to redundancy reduction; to
uniformization techniques that may explain illusions when the environment changes; to error minimization; and the
connection between visual illusions to deceiving CNNs. These papers strengthens the need to further study various
facets of the relations between image statistics, neural networks and a variety of vision/perception phenomena.

**R4:** It would help rather than hurt if the authors made that clearer (e.g., in the title, abstract, and contributions).
Thanks. We will clarify the focus of the paper, which is on color & lightness illusions, in the title & abstract. Other
types of illusions for which the empirical paradigm & setup applies (geometry, motion) are indeed left for the future.

[Meta-Review · NeurIPS 2020]

This paper provides a model to explain and generate visual illusions of certain types. Strengths include the interesting and novel problem and approach, and the potential that this line of work will lead to contributions to both computer vision and human vision research. R1, R2, R3 offer positive reviews, while R4 is concerned that the paper overclaims how general the model is, suggesting the authors make clear they are focusing on one particular type of illusion. The rebuttal promises to address this. Given the mixed reviews, the AC read the paper and agrees with R1-R3's positive opinions. Nevertheless, please address the feedback, especially R4's suggestion to more clearly and precisely state the contributions.